# Cascade RPN: Delving into High-Quality Region Proposal Network with Adaptive Convolution

**Thang Vu,     Hyunjun Jang,     Trung X. Pham,     Chang D. Yoo**
Department of Electrical Engineering
Korea Advanced Institute of Science and Technology
{thangvubk,wiseholi,trungpx,cd_yoo}@kaist.ac.kr

## Abstract

This paper considers an architecture referred to as Cascade Region Proposal Network (Cascade RPN) for improving the region-proposal quality and detection performance by *systematically* addressing the limitation of the conventional RPN that *heuristically defines* the anchors and *aligns* the features to the anchors. First, instead of using multiple anchors with predefined scales and aspect ratios, Cascade RPN relies on a *single anchor* per location and performs multi-stage refinement. Each stage is progressively more stringent in defining positive samples by starting out with an anchor-free metric followed by anchor-based metrics in the ensuing stages. Second, to attain alignment between the features and the anchors throughout the stages, *adaptive convolution* is proposed that takes the anchors in addition to the image features as its input and learns the sampled features guided by the anchors. A simple implementation of a two-stage Cascade RPN achieves AR 13.4 points higher than that of the conventional RPN, surpassing any existing region proposal methods. When adopting to Fast R-CNN and Faster R-CNN, Cascade RPN can improve the detection mAP by 3.1 and 3.5 points, respectively. The code is made publicly available at https://github.com/thangvubk/Cascade-RPN.

## 1   Introduction

Object detection has received considerable attention in recent years for its applications in autonomous driving [13, 17], robotics [3, 11] and surveillance [9, 23]. Given an image, object detectors aim to detect known object instances, each of which is assigned to a bounding box and a class label. Recent high-performing object detectors, such as Faster R-CNN [34], formulate the detection problem as a two-stage pipeline. At the first stage, a region proposal network (RPN) produces a sparse set of proposal boxes by refining and pruning a set of anchors, and at the second stage, a region-wise CNN detector (R-CNN) refines and classifies the proposals produced by RPN. Compared to R-CNN, RPN has received relatively less attention for improving its performance. This paper will focus on improving RPN by addressing its limitations that arise from heuristically defining the anchors and heuristically aligning the features to the anchors.

An anchor is defined by its scale and aspect ratio, and a set of anchors with different scales and aspect ratios are required to obtain a sufficient number of positive samples that have high overlap with the target objects. Setting appropriate scales and aspect ratios is important in achieving high detection performance, and it requires a fair amount of tuning [25, 34].

An *alignment* rule is "implicitly" defined to set up a correspondence between the image features and the reference boxes. The input features of RPN and R-CNN should be well-aligned with the bounding boxes that are to be regressed. The alignment is guaranteed in R-CNN by the RoIPool [34] or RoIAlign [18] layer . The alignment in RPN is heuristically guaranteed: the anchor boxes are *uniformly* initialized, leveraging the observation that the convolutional kernel of the RPN *uniformly*

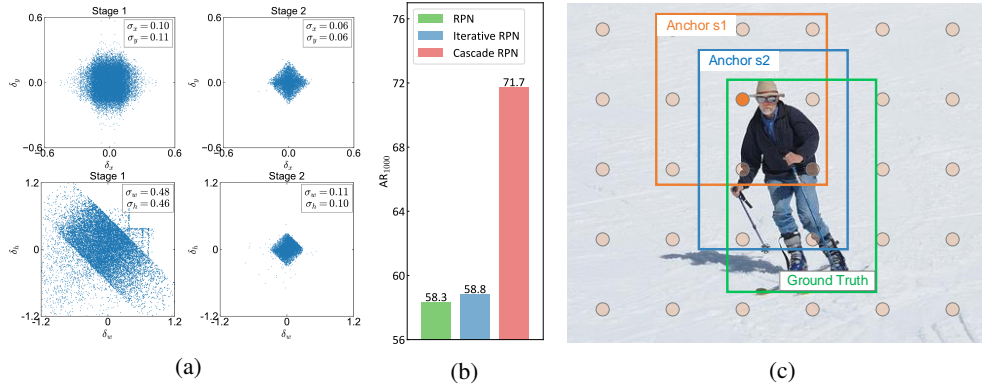

Figure 1: Iterative RPN shows limitations in improving RPN performance. (a) The target regression distribution to be learned at stage 1 and 2. The stage 2 distribution represents the error after the stage 1 distribution is learned. (b) Iterative RPN fails in learning stage-2 distribution as the average recall (AR) improvement is marginal compared to the of RPN. (c) In Iterative RPN, the anchor at stage 2, which is regressed in stage 1, breaks the alignment rule in detection.

strides over the feature maps. Such a heuristic introduces limitations for further improving detection performance as described below.

A number of studies have attempted to improve RPN by iterative refinement [14, 41]. Henceforth, this paper will refer to it as Iterative RPN. The motivation behind this idea is illustrated in Figure 1a. Anchor boxes which are references for regression are uniformly initialized, and the target ground truth boxes are arbitrarily located. Thus, RPN needs to learn a regression distribution of high variance, as shown in Figure 1a. If this regression distribution is perfectly learned, the regression distribution at stage 2 should be close to a Dirac Delta distribution. However, such a high-variance distribution at stage 1 is difficult to learn, requiring stage 2 regression. Stage 2 distribution has a lower variance compared to that of stage 1, and thus should be easier to learn but fails with Iterative RPN. The failure is implied by the observation in which the performance improvement of Iterative RPN is negligible compared to that of RPN, as shown in Figure 1b. It is explained intuitively in Figure 1c. Here, after stage 1, the anchor is regressed to be closer to the ground truth box; however, this breaks the alignment rule in detection.

This paper proposes an architecture referred to as Cascade RPN to systematically address the aforementioned problem arising from heuristically defining the anchors and aligning the features to the anchors. First, instead of using multiple anchors with different scales and aspect ratios, Cascade RPN relies on a single anchor and incorporates both anchor-based and anchor-free criteria in defining positive boxes to achieve high performance. Second, to benefit from multi-stage refinement while maintaining the alignment between anchor boxes and features, Cascade RPN relies on the proposed adaptive convolution that adapts to the refined anchors after each stage. Adaptive convolution serves as an extremely light-weight RoIAlign layer [18] to learn the features sampled within the anchors.

Cascade RPN is conceptually simple and easy to implement. Without bells and whistles, a simple two-stage Cascade RPN achieves AR 13.4 points improvement compared to RPN baseline on the COCO dataset [26], surpassing any existing region proposal methods by a large margin. Cascade RPN can also be integrated into two-stage detectors to improve detection performance. In particular, integrating Cascade RPN into Fast R-CNN and Faster R-CNN achieves 3.1 and 3.5 points mAP improvement, respectively.

## 2 Related Work

**Object Detection.** Object detection can be roughly categorized into two main streams: one-stage and two-stage detection. Here, one-stage detectors are proposed to enhance computational efficiency. Examples falling in this stream are SSD [27], YOLO [31, 32, 33], RetinaNet [25], and CornerNet [21]. Meanwhile, two-stage detectors aim to produce accurate bounding boxes, where the first stage

generates region proposals followed by region-wise refinement and classification at the second stage, *e.g.*, R-CNN [15], Fast R-CNN [16], Faster R-CNN [34], Cascade R-CNN [4], and HTC [7].

**Region Proposals.** Region proposals have become the *de-facto* paradigm for high-quality object detectors [6, 19, 20]. Region proposals serve as the attention mechanism that enables the detector to produce accurate bounding boxes while maintaining computation tractability. Early methods are based on grouping super-pixel (*e.g.*, Selective Search [36], CPMC [5], MCG [2]) and window scoring (*e.g.*, objectness in windows [1], EdgeBoxes [43]). Although these methods dominate the field of object detection in classical computer vision, they exhibit limitations as they are external modules independent of the detector and not computationally friendly. To overcome these limitations, Shaoqing *et al.* [34] propose Region Proposal Network (RPN) that shares full-image convolutional features with the detection network, enabling nearly cost-free region proposals.

**Multi-Stage RPN.** There have been a number of studies attempting to improve the performance of RPN [14, 37, 38, 41]. The general trend is to perform multi-stage refinement that takes the output of a stage as the input of the next stage and repeats until accurate localization is obtained, as presented in [14]. However, this approach ignores the problem that the regressed boxes are misaligned to the image features, breaking the alignment rule required for object detection. To alleviate this problem, recent advanced methods [12, 37] rely on deformable convolution [10] to perform feature spatial transformations and expect the learned transformations to align to the changes of anchor geometry. However, as there is no explicit supervision to learn the feature transformation, it is difficult to determine whether the improvement originates from conforming to the alignment rule or from the benefits of deformable convolution, thus making it less *interpretable*.

**Anchor-based vs. Anchor-free Criterion for Sample Discrimination.** As a bounding box usually includes an object with some amount of background, it is difficult to determine if the box is a positive or a negative sample. This problem is usually addressed by comparing the Intersection over Union (IoU) between an anchor and a ground truth box to a predefined threshold; thus, it is referred to as the anchor-based criterion. However, as the anchor is uniformly initialized, multiple anchors with different scales and aspect ratios are required at each location to ensure that there are enough positive samples [34]. The hyperparameters, such as scales and aspect ratios, are usually heuristically tuned and have a large impact on the final accuracy [25, 34]. Rather than relying on anchors, there have been studies that define positive samples by the distance between the prediction points and the center region of objects, referred to as anchor-free [35, 40, 42]. This method is simple and requires fewer hyperparameters but usually exhibits limitations in dealing with complex scenes.

## 3 Region Proposal Network and Variants

### 3.1 Region Proposal Network

Given an image $I$ of size $W \times H$, a set of anchor boxes $\mathbb{A} = \{a_{ij} \mid 0 < (i + \frac{1}{2})s \leq W, 0 < (j + \frac{1}{2})s \leq H\}$ is *uniformly* initialized over the image, with stride $s$. Unless otherwise specified, $i$ and $j$ are omitted to simplify the notation. Each anchor box $a$ is represented by a 4-tuple in the form of $a = (a_x, a_y, a_w, a_h)$, where $(a_x, a_y)$ is the center location of the anchor with the dimension of $(a_w, a_h)$. The regression branch aims to predict the transformation $\delta$ from the anchor $a$ to the target ground truth box $t$ represented as follows:

$$\begin{aligned} \delta_x &= (t_x - a_x)/a_w, & \delta_y &= (t_y - a_y)/a_h, \\ \delta_w &= \log(t_w/a_w), & \delta_h &= \log(t_h/a_h). \end{aligned} \tag{1}$$

Here, the regressor $f$ takes as input the image feature $x$ to output a prediction $\hat{\delta} = f(x)$ that minimizes the bounding box loss:

$$\mathcal{L}(\hat{\delta}, \delta) = \sum_{k \in \{x,y,w,h\}} \text{smooth}_{L_1}\left(\hat{\delta}_k - \delta_k\right), \tag{2}$$

where $\text{smooth}_{L_1}(\cdot)$ is the robust $L_1$ loss defined in [16]. The regressed anchor is simply inferred based on the inverse transformation of (1) as follows:

$$\begin{aligned} a'_x &= \hat{\delta}_x a_w + a_x, & a'_y &= \hat{\delta}_y a_h + a_y, \\ a'_w &= a_w \exp(\hat{\delta}_w), & a'_h &= a_h \exp(\hat{\delta}_h). \end{aligned} \tag{3}$$

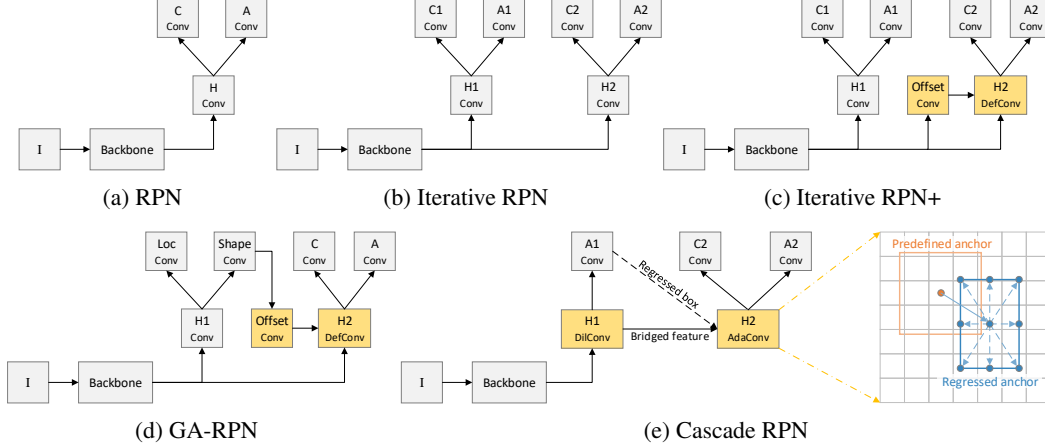

Figure 2: The architectures of different networks. "I", "H", "C", and "A" denote input image, network head, classifier, and anchor regressor, respectively. "Conv", "DefConv", "DilConv" and "AdaConv" indicate conventional convolution, deformable convolution [10], dilated convolution [39] and the proposed adaptive convolution layers, respectively.

Then the set of regressed anchor $\mathbb{A}' = \{\boldsymbol{a}'\}$ is filtered by non-maximum suppression (NMS) to produce a sparse set of proposal boxes $\mathbb{P}$:

$$\mathbb{P} = \text{NMS}(\mathbb{A}', \mathbb{S}), \tag{4}$$

where $\mathbb{S}$ is the set of objectness scores learned by the classification branch.

## 3.2 Iterative RPN and Variants

Some previous studies [14, 41] have proposed iterative refinement which is referred to as Iterative RPN, as shown in Figure 2b. Iterative RPN iteratively refines the anchors by treating $\mathbb{A}'$ as the new initial anchor set for the next stage and repeats Eqs. (1) to (3) until obtaining accurate localization. However, this approach exhibits mismatch between anchors and their represented features as the anchor positions and shapes change after each iteration.

To alleviate this problem, recent advanced methods [12, 37] use deformable convolution [10] to perform spatial transformations on the features as shown in Figure 2c and 2d and expect transformed features to align to the change in anchor geometry. However, this idea ignores the problem that there is no constraint to enforce the features to align with the changes in anchors: it is difficult to determine whether the deformable convolution produces feature transformation leading to alignment. Instead, the proposed Cascade RPN systematically ensures the alignment rule by using the proposed adaptive convolution.

## 4 Cascade RPN

### 4.1 Adaptive Convolution

Given a feature map $\boldsymbol{x}$, in the standard 2D convolution, the feature map is first sampled using a regular grid $\mathbb{R} = \{(r_x, r_y)\}$, and the samples are summed up with the weight $\boldsymbol{w}$. Here, the grid $\mathbb{R}$ is defined by the kernel size and dilation. For example, $\mathbb{R} = \{(-1, -1), (-1, 0), \ldots, (0, 1), (1, 1)\}$ corresponds to kernel size $3 \times 3$ and dilation 1. For each location $\boldsymbol{p}$ on the output feature $\boldsymbol{y}$, we have:

$$\boldsymbol{y}[\boldsymbol{p}] = \sum_{\boldsymbol{r} \in \mathbb{R}} \boldsymbol{w}[\boldsymbol{r}] \cdot \boldsymbol{x}[\boldsymbol{p} + \boldsymbol{r}]. \tag{5}$$

In adaptive convolution, the regular grid $\mathbb{R}$ is replaced by the offset field $\mathbb{O}$ that is directly inferred from the input anchor.

$$\boldsymbol{y}[\boldsymbol{p}] = \sum_{\boldsymbol{o} \in \mathbb{O}} \boldsymbol{w}[\boldsymbol{o}] \cdot \boldsymbol{x}[\boldsymbol{p} + \boldsymbol{o}]. \tag{6}$$

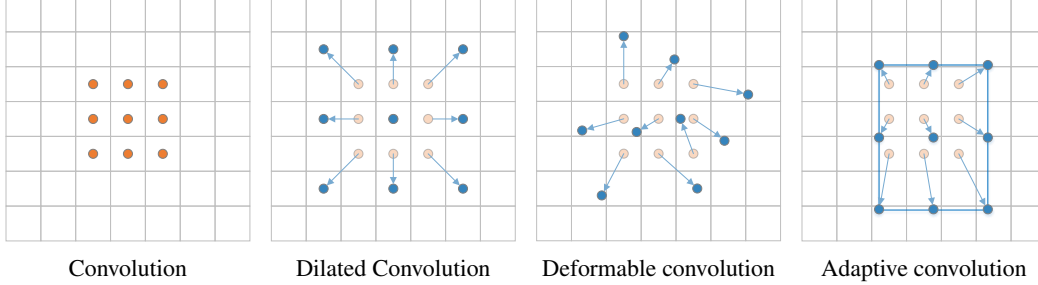

| Convolution | Dilated Convolution | Deformable convolution | Adaptive convolution |

Figure 3: Illustrations of the sampling locations in different convolutional layers with $3 \times 3$ kernel.

Let $\bar{a}$ denote the projection of anchor $\boldsymbol{a}$ onto the feature map. The offset $\boldsymbol{o}$ can be decoupled into center offset and shape offset (shown in Figure 2e):

$$\boldsymbol{o} = \boldsymbol{o}_{\text{ctr}} + \boldsymbol{o}_{\text{shp}}, \tag{7}$$

where $\boldsymbol{o}_{\text{ctr}} = (\bar{a}_x - p_x, \bar{a}_y - p_y)$ and $\boldsymbol{o}_{\text{shp}}$ is defined by the anchor shape and kernel size. For example, if kernel size is $3 \times 3$, then $\boldsymbol{o}_{\text{shp}} \in \left\{ \left(-\frac{\bar{a}_w}{2}, -\frac{\bar{a}_h}{2}\right), \left(-\frac{\bar{a}_w}{2}, 0\right), \ldots, \left(0, \frac{\bar{a}_h}{2}\right), \left(\frac{\bar{a}_w}{2}, \frac{\bar{a}_h}{2}\right) \right\}$. As the offsets are typically fractional, sampling is performed with bilinear interpolation analogous to [10].

**Relation to other Convolutions.** The illustrations of sampling locations in adaptive and other related convolutions are shown in Figure 3. Conventional convolution samples the features at contiguous locations with a dilation factor of 1. The dilated convolution [39] increases the dilation factor, aiming to enhance the semantic scope with unchanged computational cost. The deformable convolution [10] augments the spatial sampling locations by learning the offsets. Meanwhile, the proposed adaptive convolution performs sampling within the anchors to ensure alignment between the anchors and features. Adaptive convolution is closely related to the others. Adaptive convolution becomes dilated convolution if the center offsets are zeros. Deformable convolution becomes adaptive convolution if the offsets are deterministically derived from the anchors.

## 4.2 Sample Discrimination Metrics

Instead of using multiple anchors with predefined scales and aspect ratios, Cascade RPN relies on a single anchor per location and performs multi-stage refinement. However, this reliance creates a new challenge in determining whether a training sample is positive or negative as the use of anchor-free or anchor-based metric is highly adversarial. The anchor-free metric establishes a loose requirement for positive samples in the second stage and the anchor-based metric results in an insufficient number of positive training examples at the first stage. To overcome this challenge, Cascade RPN progressively strengthens the requirements through the stages by starting out with an anchor-free metric followed by anchor-based metrics in the ensuing stages. In particular, at the first stage, an anchor is a positive sample if its center is inside the center region of an object. In the following stages, an anchor is a positive sample if its IoU with an object is greater than the IoU threshold.

## 4.3 Cascade RPN

The architecture of a two-stage Cascade RPN is illustrated in Figure 2e. Here, Cascade RPN relies on adaptive convolution to systematically align the features to the anchors. In the first stage, the adaptive convolution is set to perform dilated convolution since anchor center offsets are zeros. The features of the first stage are "bridged" to the next stages since the spatial order of the features is maintained by the dilated convolution. The pipeline of the proposed Cascade RPN can be described mathematically in Algorithm 1. The anchor set at the first stage $\mathbb{A}^1$ is uniformly initialized over the image. At stage $\tau$, the anchor offset $\boldsymbol{o}^\tau$ is computed and fed into the regressor $f^\tau$ to produce the regression prediction $\hat{\boldsymbol{\delta}}^\tau$. The prediction $\hat{\boldsymbol{\delta}}^\tau$ is used to produce regressed anchors $\boldsymbol{a}^{\tau+1}$. At the final stage, the objectness scores are derived from the classifier, followed by NMS to produce the region proposals.

**Algorithm 1.** Cascade RPN
___
1 **Input**: sequence of regressors $f^\tau$, classifier $g$, feature $\boldsymbol{x}$ of image $I$.
2 **Output**: proposal set $\mathbb{P}$.
3 Uniformly initialize anchor set $\mathbb{A}^1 = \{\boldsymbol{a}^1\}$ over image $I$.
4 **for** $\tau \leftarrow 1$ **to** $T$ **do**
5      Compute offset $\boldsymbol{o}^\tau$ of input anchor $\boldsymbol{a}^\tau$ on feature map using (7).
6      Compute regression prediction $\hat{\boldsymbol{\delta}}^\tau = f^\tau(\boldsymbol{x}, \boldsymbol{o}^\tau)$.
7      Compute regressed anchor $\boldsymbol{a}^{\tau+1}$ from $\hat{\boldsymbol{\delta}}^\tau$ using (3).
8 **end**
9 Compute objectness score $\boldsymbol{s} = g(\boldsymbol{x}, \boldsymbol{o}^T)$.
10 Derive proposals $\mathbb{P}$ from $\mathbb{A}^{\tau+1} = \{\boldsymbol{a}^{\tau+1}\}$ and $\mathbb{S} = \{\boldsymbol{s}\}$ using NMS (4).
___

### 4.4 Learning

Cascade RPN can be trained in an end-to-end manner using multi-task loss as follows:

$$\mathcal{L} = \lambda \sum_{\tau=1}^{T} \alpha^\tau \mathcal{L}_{reg}^\tau + \mathcal{L}_{cls}. \tag{8}$$

Here, $\mathcal{L}_{reg}^\tau$ is the regression loss at stage $\tau$ with the weight of $\alpha^\tau$, and $\mathcal{L}_{cls}$ is the classification loss. The two loss terms are balanced by $\lambda$. In the implementation, binary cross entropy loss and IoU loss [40] are used as the classification loss and regression loss, respectively.

## 5 Experiments

### 5.1 Experimental Setting

The experiments are performed on the COCO 2017 detection dataset [26]. All the models are trained on the `train` split (115k images). The region proposal performance and ablation analysis are reported on `val` split (5k images), and the benchmarking detection performance is reported on `test-dev` split (20k images).

Unless otherwise specified, the default model of the experiment is as follows. The model consists of two stages, with ResNet50-FPN [24] being its backbone. The use of two stages is to balance accuracy and computational efficiency. A single anchor per location is used with size of $32^2$, $64^2$, $128^2$, $256^2$, and $512^2$ corresponding to the feature levels $C_2$, $C_3$, $C_4$, $C_5$, and $C_6$, respectively [24]. The first stage uses the anchor-free metric for sample discrimination with the thresholds of the center-region $\sigma_{\text{ctr}}$ and ignore-region $\sigma_{\text{ign}}$, which are adopted from [40, 37], being 0.2 and 0.5. The second stage uses the anchor-based metric with the IoU threshold of 0.7. The multi-task loss is set with the stage-wise weight $\alpha^1 = \alpha^2 = 1$ and the balance term $\lambda = 10$. The NMS threshold is set to 0.8. In all experiments, the long edge and the short edge of the images are resized to 1333 and 800 respectively without changing the aspect ratio. No data augmentation is used except for standard horizontal image flipping. The models are implemented with PyTorch [29] and mmdetection [8]. The models are trained with 8 GPUs with a batch size of 16 (two images per GPU) for 12 epochs using SGD optimizer. The learning rate is initialized to 0.02 and divided by 10 after 8 and 11 epochs. It takes about 12 hours for the models to converge on 8 Tesla V100 GPUs.

The quality of region proposals is measured with Average Recall (AR), which is the average of recalls across IoU thresholds from 0.5 to 0.95 with a step of 0.05. The AR for 100, 300, and 1000 proposals per image are denoted as $\text{AR}_{100}$, $\text{AR}_{300}$, and $\text{AR}_{1000}$. The AR for small, medium, and large objects computed at 100 proposals are denoted as $\text{AR}_S$, $\text{AR}_M$, and $\text{AR}_L$, respectively. Detection results are evaluated with the standard COCO-style Average Precision (AP) measured at IoUs from 0.5 to 0.95. The runtime is measured on a single Tesla V100 GPU.

Table 1: Region proposal results on COCO 2017 `val`.

| Method | Backbone | $AR_{100}$ | $AR_{300}$ | $AR_{1000}$ | $AR_S$ | $AR_M$ | $AR_L$ | Time (s) |
|---|---|---|---|---|---|---|---|---|
| SharpMask [30] | ResNet-50 | 36.4 | - | 48.2 | - | - | - | 0.76 |
| GCN-NS [28] | VGG-16 (Sync BN) | 31.6 | - | 60.7 | - | - | - | 0.10 |
| AttractioNet [14] | VGG-16 | 53.3 | - | 66.2 | 31.5 | 62.2 | 77.7 | 4.00 |
| ZIP [22] | BN-inception | 53.9 | - | 67.0 | 31.9 | 63.0 | 78.5 | 1.13 |
| RPN [34] | | 44.6 | 52.9 | 58.3 | 29.5 | 51.7 | 61.4 | **0.04** |
| Iterative RPN | | 48.5 | 55.4 | 58.8 | 32.1 | 56.9 | 65.4 | 0.05 |
| Iterative RPN+ | ResNet-50-FPN | 54.0 | 60.4 | 63.0 | 35.6 | 62.7 | 73.9 | 0.06 |
| GA-RPN [37] | | 59.1 | 65.1 | 68.5 | 40.7 | 68.2 | 78.4 | 0.06 |
| Cascade RPN | | **61.1** | **67.6** | **71.7** | **42.1** | **69.3** | **82.8** | 0.06 |

Table 2: Detection results on COCO 2017 `test-dev`

| Method | Proposal method | # proposals | AP | $AP_{50}$ | $AP_{75}$ | $AP_S$ | $AP_M$ | $AP_L$ |
|---|---|---|---|---|---|---|---|---|
| Fast R-CNN | RPN | 1000 | 37.0 | **59.5** | 39.9 | 21.1 | 39.4 | 47.0 |
| | Cascade RPN | | **40.1** | **59.5** | **43.7** | **22.8** | **42.4** | **50.9** |
| | RPN | | 36.6 | 58.6 | 39.5 | 20.3 | 39.1 | 47.0 |
| | Iterative RPN+ | 300 | 38.6 | 58.8 | 42.2 | 21.1 | 41.5 | 50.0 |
| | GA-RPN | | 39.5 | 59.3 | 43.2 | 21.8 | 42.0 | 50.7 |
| | Cascade RPN | | **40.1** | **59.4** | **43.8** | **22.1** | **42.4** | **51.6** |
| Faster R-CNN | RPN | 1000 | 37.1 | **59.3** | 40.1 | 21.4 | 39.8 | 46.5 |
| | Cascade RPN | | **40.5** | **59.3** | **44.2** | **22.6** | **42.9** | **51.5** |
| | RPN | | 36.9 | 58.9 | 39.9 | 21.1 | 39.6 | 46.5 |
| | Iterative RPN+ | 300 | 39.2 | 58.2 | 43.0 | 21.5 | 42.0 | 50.4 |
| | GA-RPN | | 39.9 | **59.4** | 43.6 | **22.0** | 42.6 | 50.9 |
| | Cascade RPN | | **40.6** | 58.9 | **44.5** | **22.0** | **42.8** | **52.6** |

## 5.2 Benchmarking Results

**Region Proposal Performance.** The performance of Cascade RPN is compared to those of recent state-of-the-art region proposal methods, including RPN [34], SharpMask [30], GCN-NS [28], AttractioNet [14], ZIP [22], and GA-RPN [37]. In addition, Iterative RPN and Iterative RPN+, which are referred to in Figure 2, are also benchmarked. The results of Sharp Mask, GCN-NS, AttractioNet, ZIP are cited from the papers. The results of the remaining methods are reproduced using mmdetection [8]. Table 1 summarizes the benchmarking results. In particular, Cascade RPN achieves AR 13.4 points higher than that of the conventional RPN. Cascade RPN consistently outperforms the other methods in terms of AR under different settings of proposal thresholds and object scales. The alignment rule is typically missing or loosely conformed to in the other methods; thus, their performance improvements are limited. The alignment rule in Cascade RPN is systematically ensured such that the performance gain is greater and more reliable.

**Detection Performance.** To investigate the benefit of high-quality proposals, Cascade RPN and the baselines are integrated into common two-stage object detectors, including Fast R-CNN and Faster R-CNN. Here, Fast R-CNN is trained on precomputed region proposals while Faster R-CNN is trained in an end-to-end manner. As studied in [37], despite high-quality region proposals, training a good detector is still a non-trivial problem, and simply replacing RPN by Cascade RPN without changes in the settings only brings limited gain. Following [37], the IoU threshold in R-CNN is increased and the number of proposals is decreased. In particular, the IoU threshold and the number of proposals are set to 0.65 and 300, respectively. The experimental results are reported in Table 2. Here, integrating RPN into Fast R-CNN and Faster R-CNN yields 37.0 and 37.1 mAP, respectively. From the results, the recall improvement is correlated with improvements in detection performance. As it has the highest recall, Cascade RPN boosts the performance for Fast R-CNN and Faster R-CNN to 40.1 and 40.6 mAP, respectively.

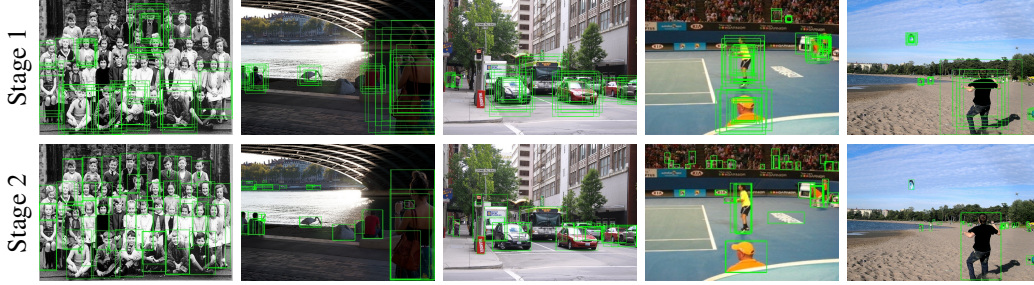

Figure 4: Examples of region proposal results at stage 1 (first row) and stage 2 (second row) of Cascade RPN.

Table 3: Ablation analysis of Cascade RPN. Here, Align., AFAB, and Stats. denote the use of alignments, anchor-free and anchor-based metrics, and regression statistics, respectively.

| Baseline | 1 anchor | Cascade | Align. | AFAB | Stats. | IoU loss | $AR_{100}$ | $AR_{300}$ | $AR_{1000}$ |
|---|---|---|---|---|---|---|---|---|---|
| ✓ | | | | | | | 44.6 | 52.9 | 58.3 |
| | ✓ | | | | | | 44.7 | 51.2 | 55.8 |
| | ✓ | ✓ | | | | | 48.2 | 54.4 | 58.0 |
| | ✓ | ✓ | ✓ | | | | 57.4 | 63.7 | 67.8 |
| | ✓ | ✓ | ✓ | ✓ | | | 57.3 | 64.2 | 68.6 |
| | ✓ | ✓ | ✓ | ✓ | ✓ | | 60.8 | 67.3 | 71.5 |
| | ✓ | ✓ | ✓ | ✓ | ✓ | ✓ | **61.1** | **67.6** | **71.7** |
| Overall Improvement | | | | | | | **+16.5** | **+14.7** | **+13.4** |

Table 4: The effects of alignment

| Center | Shape | $AR_{100}$ | $AR_{300}$ | $AR_{1000}$ |
|---|---|---|---|---|
| | | 48.2 | 54.4 | 58.0 |
| ✓ | | 52.5 | 59.4 | 64.1 |
| ✓ | ✓ | **57.4** | **63.7** | **67.8** |

Table 5: The effects of sample metrics

| AF | AB | $AR_{100}$ | $AR_{300}$ | $AR_{1000}$ |
|---|---|---|---|---|
| ✓ | | 55.2 | 61.8 | 66.4 |
| | ✓ | **57.4** | 63.7 | 67.8 |
| ✓ | ✓ | 57.3 | **64.2** | **68.6** |

## 5.3 Ablation Study

**Component-wise Analysis.** To demonstrate the effectiveness of Cascade RPN, a comprehensive component-wise analysis is performed in which different components are omitted. The results are reported in Table 3. Here, the baseline is RPN with 3 anchors per location yielding $AR_{1000}$ of 58.3. When the number of anchors per location is reduced to 1, the $AR_{1000}$ drops to 55.8, implying that the number of positive samples dramatically decreases. Even when the multi-stage cascade is added, the performance is 58.0, which is still lower than that of the baseline. However, when adaptive convolution is applied to ensure alignment, the performance surges to 67.8, showing the importance of alignment in multi-stage refinement. The incorporation of anchor-free and anchor-based metrics for sample discrimination incrementally improves $AR_{1000}$ to 68.6. The use of regression statistics (shown in Figure 1a) increases the performance to 71.5. Finally, applying IoU loss yields a slight improvement of 0.2 points. Overall, Cascade RPN achieves 16.5, 14.7, and 13.4 points improvement in terms of $AR_{100}$, $AR_{300}$, and $AR_{1000}$ respectively, compared to the conventional RPN.

**Acquisition of Alignment.** To demonstrate the effectiveness of the proposed adaptive convolution, the center and shape alignments, represented by the offsets in Eq. (7), are progressively applied. Here, the center and shape offsets maintain the alignments in position and semantic scope, respectively. Table 4 shows that the $AR_{1000}$ improves from 58.0 to 64.1 using only the center alignment. When both the center and shape alignments are ensured, the performance increases to 67.8.

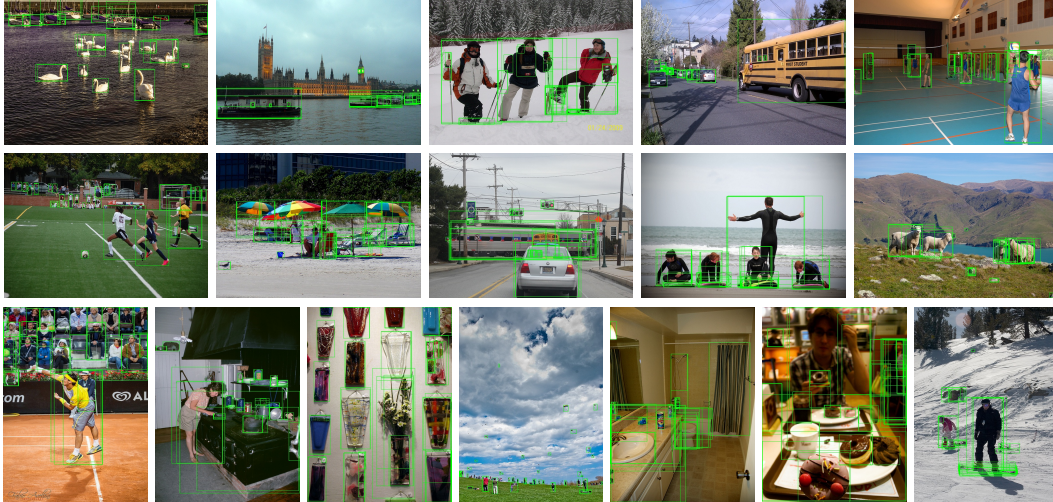

Figure 5: More examples of region proposal results of Cascade RPN.

Table 6: Ablation study on # stages.

| # stages | $AR_{100}$ | $AR_{300}$ | $AR_{1000}$ | Time (s) |
|---|---|---|---|---|
| 1 | 56.0 | 62.2 | 66.3 | **0.04** |
| 2 | **61.1** | 67.6 | 71.7 | 0.06 |
| 3 | 60.9 | **67.9** | **72.2** | 0.08 |

Table 7: Detection results of Cascade R-CNN with RPN and Cascade RPN (denoted by CRPN).

| Method | AP | $AP_{50}$ | $AP_{75}$ | $AP_{75}$ | $AP_S$ | $AP_M$ |
|---|---|---|---|---|---|---|
| RPN | 40.8 | **59.3** | 44.3 | 22.0 | 44.2 | 54.2 |
| CRPN | **41.6** | 59.0 | **45.5** | **23.0** | **45.0** | **55.2** |

**Sample Discrimination Metrics.** The experimental results with different combinations of sample discrimination metrics are shown in Table 5. Here, AF and AB denote that the anchor-free and anchor-based metrics are applied for all stages, respectively. Meanwhile, AFAB indicates that the anchor-free metric is applied at stage 1 followed by anchor-based metric at stage 2. Here, AF and AB yield the $AR_{1000}$ of 66.4 and 67.8 respectively, both of which are significantly less than that of AFAB. It is noted that the thresholds for each metric are already adapted through stages. The results imply that applying only one of either anchor-free or anchor-based metric is highly adversarial. The both metrics should be incorporated to achieve the best results.

**Qualitative Evaluation.** The examples of region proposal results at the first and second stages are illustrated in the first and second row of Figure 4, respectively. The results show that the output proposals at the second stage are more accurate and cover a larger number of objects.

**Number of Stages.** Table 6 shows the proposal performance on different number of stages. In the 3-stage Cascade RPN, an IoU threshold of 0.75 is used for the third stage. The 2-stage Cascade RPN achieves the best trade-off between $AR_{1000}$ and inference time.

**Extension with Cascade R-CNN.** Table 7 reports the detection results of the Cascade R-CNN [4] with different proposal methods. The Cascade RPN improves AP by 0.8 points compared to RPN. The improvement is mainly from $AP_{75}$, where the objects have high IoU with the ground truth.

# 6 Conclusion

This paper introduces Cascade RPN, a simple yet effective network architecture for improving region proposal quality and object detection performance. Cascade RPN systematically addresses the limitations that conventional RPN heuristically defines the anchors and aligns the features to the anchors. A simple implementation of a two-stage Cascade RPN achieves AR 13.4 points higher than the baseline, surpassing any existing region proposal methods. When adopting to Fast R-CNN and Faster R-CNN, Cascade RPN can improve the detection mAP by 3.1 and 3.5 points, respectively.

**Acknowledgment.** This work was supported by Institute for Information & communications Technology Planning & Evaluation(IITP) grant funded by the Korea government (MSIT) (2017-0-01780, The technology development for event recognition/relational reasoning and learning knowledge-based system for video understanding) and (No. 2019-0-01396, Development of framework for analyzing, detecting, mitigating of bias in AI model and training data)

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
