[Reviews · NeurIPS 2019]

Reviewer 1



This paper is about Cascade RPN, a novel region proposal network which is typically used to propose potential boxes containing objects of an image. Starting from the observation that the original RPN has fixed anchors distributed uniformly and do not reflect the arbitrary locations of ground truth boxes, the paper describes the recent attempts to address this issue. Accurate description of Iterative RPN, its plus version, and GA-RPN is reported with their issues regarding the feature alignment problem. The paper then describes the use of adaptive convolution and the use of iterative anchor distribution which form the base of Cascade RPN. Anchors are predicted dynamically along with their features properly aligned. Experiments on the popular COCO2017 detection dataset show that the proposed Cascade RPN has better performance vs previous RPN models in terms of average recall. The integration of Cascade RPN with state of the art object detection model Fast RCNN and Faster RCNN show a slight improvement on mAP performance. Strengths: + The paper is very well written, clear and easy to follow. + The proposed Cascade RPN goes into the good direction of improving the RPN network by dynamically set the anchors. + The method is sound and experiments seems to be well carried. + Performance in terms of average recall is clearly improved by the approach, and the integration of the model into Fast(er)-RCNN show a little improvement. Weaknesses: - The approach is a refinement of previous methods (iterative RPN, GA-RPN) and thus incremental. Nonetheless, the performance are significant. - More analysis regarding the various parameters would have been interesting. For instance, typically a region proposal is tested according to its average recall by varying the number of proposal emitted. It would have been interesting to see a plot when varying such number. After the rebuttal update —————————— I thank the authors for addressing my concerns. As a result, I increase my score to accept.

Reviewer 2



The paper proposes a new variant of the popular region proposal networks (RPN). The authors first identify the issue stemming from a misalignment between predefined anchors and the ground truth boxes. They investigate why this issue is not resolved well enough by iterative RPNs and propose to address the issue directly by removing the heuristically defined anchors. This is enabled by: - using a single box per position, - using a combination of anchor-free and anchor-based criteria to define the positive boxes, - introducing a new adaptive convolution layer that allows the features to be well-aligned with the anchors. The experiments show that the method clearly surpasses other state-of-the-art methods (2-4% over the 2nd best method GA-RPN in about the same runtime). The authors also perform an ablation study which shows the contribution of the individual components. Overall the paper is clearly written and well structured. Questions / Changes: - I wonder why the improvements seem to be larger for mainly for large objects (Table 2, AP_L). Is there an explanation? - Why do you prefer the IOU loss over the more common L1-smooth loss? Have you tested both? - Please make the font in Figs. 1 & 2 larger - as of now it is too small, which makes it hard to read the labels. - Correct the typo on L172 "object objectness" -> "objectness". Also after the rebuttal I think this paper should be accepted.

Reviewer 3



The idea is simple and valid, and the experimental results demonstrate improvement over other methods. The writing of the paper is also good, with strong logic and easy to follow. My major uncertainty, is whether cascade-rpn could be plug in other popular two-stage object detection approaches, and achieve better object detection results.

[Author Response · NeurIPS 2019]

Table 1: Detection results of Faster R-CNN with varying # proposals

| # proposals | AP | |
| --- | --- | --- |
| | w/ RPN | w/ CRPN |
| 100 | 34.8 | 38.2 |
| 300 | 37.0 | 40.5 |
| 1000 | 36.8 | 40.4 |

Table 2: Detection results of Cascade R-CNN w.r.t. different RPN methods

| Method | Proposal method | AP |
| --- | --- | --- |
| Cascade R-CNN | RPN | 40.4 |
| | CRPN | 41.2 |

Table 3: Proposal results with different # stages

| # stages | $AR_{1000}$ | Time (s) |
| --- | --- | --- |
| 1 | 66.3 | 0.04 |
| 2 | 71.7 | 0.06 |
| 3 | 72.2 | 0.08 |

Figure 1: Examples of region proposal results at stage 1 (first row) and stage 2 (second row) of Cascade RPN

We thank the reviewers for their valuable feedback and encouraging comments.

**1. Response to Reviewer #1**

*1.1. Performance with varying the number of proposals.* If we understand the question correctly, the reviewer is probably more concerned with AP as opposed to AR of different RPN methods since the AR w.r.t. different number of proposals was presented in the Table 1 of the paper. Here, we report the AP of Faster R-CNN on COCO 2017 `val` with 100, 300, and 1000 proposals in Table 1, showing that Cascade RPN consistently outperforms conventional RPN.

**2. Response to Reviewer #2**

*2.1. Higher AP improvements for large objects.* The Cascade RPN does not have preferences for large objects as it does not make any prior assumptions regarding the object scale. Large objects are simply easier to detect than small objects, and the difference in performance gain w.r.t. object scales is also observed in other RPN methods such as the GA-RPN.

*2.2. Why IOU loss?* The use of IoU loss is simply a design choice. The IoU loss regresses the bounding box as a single unit instead of 4 independent variables considered in L1-smooth loss. The experiment results when using the different losses were performed and reported Table 3 of the paper: IoU loss marginally improves $AR_{1000}$ from 71.5 to 71.7.

*2.3. Figure label size and typos.* We will revise and fix the label size and typos.

**3. Response to Reviewer #3**

*3.1. Performance with Cascade R-CNN.* Table 2 shows preliminary results obtained using Cascade R-CNN which is fine-tuned with precomputed proposals acquired from converged Cascade RPN. Here, the hyper-parameters of the Cascade R-CNN are not modified for fine-tuning. The Cascade RPN improves AP by 0.8 points compared to RPN. This performance could probably be further improved with a better hyper-parameter setting of Cascade R-CNN.

*3.2. Performance w.r.t. different number of stages.* These experiment results are shown in Table 3. In 3-stage Cascade RPN, an IoU threshold of 0.75 is used for the 3rd stage. The 2-stage Cascade RPN achieves the best trade-off between $AR_{1000}$ and inference time. Increasing # stages to 3 improves $AR_{1000}$ by 0.5 points but results in 33% slower.

*3.3. Using multiple anchor shapes.* We have not used or tried multiple anchor shapes per location. The main theme of the paper is to avoid heuristically defined hyper-parameters such as aspect-ratios of anchors. However, Cascade RPN can be easily extended to multiple anchor shapes.

*3.4. Visualization at different stages.* The output proposals of stage 1 and 2 of Cascade RPN are shown respectively in first and second rows of Figure 1. The proposals at stage 2 are more accurate and cover a larger number of objects.

*3.5. Motivation of anchor-free metric for the first stage.* The motivations are twofold: (1) the anchor-free metric is more relaxed than anchor-based metric since anchor-free metric is defined irrespective of the anchor shape and (2) at first stage, anchors are initialized uniformly over the image, using anchor-based metric may result in poor overlaps between anchors and ground truth boxes. We reported the detailed ablation study on sample metrics in Table 5 in the paper. The best $AR_{1000}$ we were able to obtain when using anchor-based metric is 67.8, where the IoU thresholds of 0.5 and 0.7 were used in the first and second stages. Combining anchor-free and anchor-based metric improves $AR_{1000}$ to 68.6.

*3.6. "CRAFT Objects from Images" paper.* We thank the reviewer for the constructive suggestion. We will cite the CRAFT method in our paper.

[Meta-Review · NeurIPS 2019]

The three reviewers agreed on the strength of the contribution and praised both the interest of the method presented and the displayed performance. The rebuttal also provides strong arguments in favor of the approach with additional experiments which offer additional insights. I warmly recommend the paper for a spotlight presentation during the conference.